# Carers’ Perspective on Voluntary Stopping of Eating and Drinking: A Systematic Mixed-Methods Review of Motives and Attitudes

**DOI:** 10.3390/healthcare13111264

**Published:** 2025-05-27

**Authors:** Christina Mensger, Julien Poehner, Maximiliane Jansky, Yang Jiao, Friedemann Nauck, Henrikje Stanze

**Affiliations:** 1Institute of Public Health, Charité—Universitätsmedizin Berlin, Charitéplatz 1, 10117 Berlin, Germany; yang.jiao@posteo.de; 2Centre for Nursing Research and Counselling, School of Social Science, Hochschule Bremen—City University of Applied Sciences, Am Brill 2–4, 28195 Bremen, Germany; julien.poehner@hs-bremen.de (J.P.); henrikje.stanze@hs-bremen.de (H.S.); 3Department of Palliative Medicine, University Medical Center Göttingen, Georg-August University of Göttingen, Robert-Koch-Straße 40, 37075 Göttingen, Germany; maximiliane.jansky@med.uni-goettingen.de (M.J.); friedemann.nauck@med.uni-goettingen.de (F.N.)

**Keywords:** voluntary stopping of eating and drinking, refusal of food and liquid, systematic review, hasten death, end-of-life, health personnel, family, attitude, experience

## Abstract

**Background/Objectives**: Voluntary stopping of eating and drinking (VSED) is a way to hasten death or end life prematurely. VSED is controversial, and research is essential to inform this debate. The aim was to systematically synthesize empirical data on the attitudes, motives, and experiences related to VSED from the perspective of caregivers. **Methods**: This systematic mixed-methods review (PROSPERO CRD42022283743) included qualitative and quantitative research. The MEDLINE, CINAHL, PsycINFO, Google Scholar, and BELIT databases were searched for English and German articles published between 1 January 2013 and 12 November 2021. Studies examining experiences, attitudes, and knowledge regarding VSED were eligible. We analyzed the data inductively after quantitative data transformation. Quality and confidence were assessed using the Mixed Methods Appraisal Tool (MMAT) and GRADE-CERQual approach, respectively. **Results**: We identified 22 articles, including 16 studies. The participants were healthcare professionals and relatives, but not those who chose VSED. The motives for VSED are based on high symptom burdens and are closely related to self-determination. Most perceive VSED as a natural death and accept the patient’s decision. However, this acceptance also depends on other factors, such as patient characteristics. Most healthcare professionals would accompany patients during VSED, sometimes leading to advocacy. Relatives often play an active role in VSED and may experience distressing grief if they struggle with their support or the dying situation. The confidence in the review findings ranged from moderate to very low. Our findings on the challenging and positive experiences related to VSED and the resulting needs have been published elsewhere. **Conclusions**: VSED is a complex phenomenon affecting patients, nursing staff, physicians, and relatives. We provide an empirical basis for VSED to support research, debate, and practice.

## 1. Introduction

Individuals persistently wishing to die sometimes consciously abstain from drinking and eating to hasten their death or end their lives prematurely, known as voluntary stopping of eating and drinking (VSED) [1].

Aging populations have led to more frequent encounters with chronic and terminal illnesses, prompting discussions about dignified death. Additionally, self-determination and autonomy are increasingly recognized as significant societal trends in many countries, influencing end-of-life decision-making. Ongoing debates concerning laws on medically assisted dying further underscore the relevance on these end-of-life decisions. In this context, voluntary stopping of eating and drinking (VSED) has become an emerging focus of public and scientific attention, as patients and healthcare professionals must navigate complex ethical and legal landscapes.

In addition to VSED, other terms include voluntary refusal of food and fluids (VRFF) [2], patient refusal of hydration and nutrition (PRHN) [3], voluntary palliated starvation [4], or terminal dehydration [5]. There may be slightly different definitions between these terms. For example, Savulescu argues that every competent person has the right to refuse food and drink, even if this leads to death. Considering that every person is entitled to medical relief of their suffering through palliative care, which may include sedation and analgesia, he refers to this combination of refusal of food and drink and palliative symptom control as “voluntary palliative starvation” [4]. Terminal dehydration has only been described in the context of terminally ill patients and includes the decision to forego artificial nutrition and hydration [5].

Voluntary stopping of eating and drinking (VSED) refers to patients without cognitive impairment or mental illness [1]. The decision for VSED must be distinguished from other forms of food refusal, such as natural anorexia, the physical inability to ingest or digest oral food, or the naturally occurring loss of appetite and interest in eating and drinking that often accompanies the final stages of dying [6,7].

Although the decision to begin VSED and the continuous choice to refrain from eating and drinking are personal, this process necessitates substantial social (and practical) support as the individual becomes progressively weaker and less able to care for themselves [7,8]. Death by stopping eating and drinking has been described to occur after approximately one to two weeks. Nonetheless, the course may vary depending on the underlying disease conditions, weakness, and fluid intake [9,10]. The most common symptoms in the last three days before death reported by healthcare professionals were pain, fatigue, impaired cognitive abilities, and thirst or a dry throat [10,11]. The symptom burden and discomfort can be managed and alleviated by palliative care measures [8,12]. It is not unlikely that healthcare professionals in nursing or palliative care will come into contact with the phenomenon of VSED. Studies show that 32–62% of primary or home care healthcare professionals had already accompanied a person during VSED [10,13,14,15,16,17,18].

VSED is a controversial topic; many reviews and comments have been published addressing ethical and legal issues regarding VSED [7,8,19,20,21]. In these publications, the question of whether VSED should be classified as suicide was the most critical point of the debate that was widely discussed [7,9,22,23,24,25,26,27,28,29,30,31,32,33,34]. Case reports [35,36,37,38], books based on case reports [39,40,41], and personal narratives of individuals who accompanied a loved one during VSED have been published [37,42,43].

The first synthesis of empirical research on VSED was conducted in 2013 through a mapping review [44], followed by a systematic search and review in 2014 [1]. This systematic review included sixteen articles, comprising four surveys [2,45,46,47], four case reports, and eight narrative reviews on VSED. The authors summarized that these articles provide only a marginal insight into VSED and concluded that further research is needed from different perspectives (patients, family members, and healthcare professionals), primarily through qualitative studies. Since then, several empirical studies have been conducted, making it necessary to update the evidence synthesis.

Objective: This systematic review aimed to synthesize all available data from empirical studies conducted since 2013 on the experiences, knowledge, and attitudes of healthcare professionals, relatives, close friends, and affected persons towards VSED. This article presents our findings on motives, decision-making, attitudes, and advocacy during accompaniment from caregivers’ perspectives. Our findings on the challenges and positive experiences related to VSED and the resulting needs have been published elsewhere [48].

## 2. Materials and Methods

This systematic review follows the PRISMA [49] (Appendix A) and EPOC [50] guidelines. A protocol was registered in PROSPERO (CRD42022283743; https://www.crd.york.ac.uk/PROSPERO/view/CRD42022283743) on 8 January 2022. The PRISMA flowchart, the search strategy, including search terms and a more detailed description of the methods on data analysis, quality assessment, and confidence assessment, are published in Mensger et al., 2024 and the corresponding Supplementary Material [48].

### 2.1. Design and Selection Criteria

This systematic mixed-methods review searched for empirical studies using qualitative or quantitative data collection and analysis methods and mixed-methods studies. Case reports and case series were excluded. We were interested in the experiences and attitudes of people confronted with the voluntary cessation of eating and drinking and their knowledge of this phenomenon. Study participants could be healthcare professionals, relatives, individuals who have chosen VSED themselves, or others who have been confronted with VSED or could report on the perspectives of healthcare professionals, relatives, or those affected. Studies were included with all types of VSED and were not restricted to patients with terminal illnesses. Studies on people who stopped eating and drinking for other reasons (e.g., anorexia nervosa or political reasons such as hunger strikes) were excluded. We included studies conducted in any country and published in English or German between 2013 and 2021.

### 2.2. Search and Data Analysis

The main search was conducted on 12 November 2021 in the Medline, CINAHL, and PsychINFO databases, supplemented by Google Scholar and BELIT searches. We analyzed data using a convergent integrated approach recommended by Joanna Briggs Guidance in case the research question could be answered by quantitative and qualitative research [51]. This approach requires data transformation, and we chose to “qualify” the quantitative data because our research question is qualitative, and the application of the CERQual approach (see next paragraph) requires the creation of qualitative results (review findings). We then used an adapted version of Thomas and Harden’s “thematic synthesis” to inductively synthesize the data and produce qualitative review findings [52].

### 2.3. Assessing Confidence in the Review Findings

We applied the GRADE-CERQual (Confidence in the Evidence from Reviews of Qualitative research) approach to assess confidence in the individual review findings. Confidence in the evidence “is an assessment of the extent to which a review finding is a reasonable representation of the phenomenon of interest” [53] (p. 1). The CERQual approach comprises four components: methodological limitations [54], coherence [55], adequacy [56], and relevance [57]. Consideration of each component within this structured framework ultimately leads to an overall confidence assessment for each review finding [58]. We assessed the methodological limitations of the qualitative, quantitative, and mixed-method studies using the Mixed Methods Appraisal Tool (MMAT) [59]. The CERQual assessments for all review findings presented here were conducted independently by two authors (C.M. and J.P.). All disagreements were discussed until a consensus was reached. Notably, the CERQual assessment is not a judgment of the work of the study authors but provides an assessment of confidence in the evidence at the time. These concerns can help identify and highlight research gaps. Notably, in this systematic review, the CERQual approach was adapted for application to both qualitative and quantitative research. It should be noted that the CERQual approach is officially recommended only for qualitative studies. For further details regarding this adaptation, please refer to the Supplementary Material of Mensger et al. [48].

## 3. Results

The systematic search identified 22 publications describing 16 studies (Table 1). Three publications were dissertations that supplemented a corresponding publication [60,61,62]. At the time of this systematic search, no studies involving individuals who decided to undergo VSED were identified. Further details on the characteristics of the included studies have been published elsewhere [48].

### 3.1. Review Findings

In this article, we present 10 review findings on the motives, decision-making, attitudes, and acceptance of VSED, advocacy during accompaniment, and the grief of relatives following the loss of a loved one through VSED (Table 2). The results, which focus on positive and challenging experiences and the resulting needs related to VSED, have been published elsewhere [48] (Appendix A). A structured summary of all 10 review findings presented here is also included in the CERQual assessment table (Table 3).

Motives—high symptom burden and suffering (reported by HCPs and relatives):

Persons who have decided on VSED typically had a high symptom burden of physical and psychological symptoms, were in poor health, or had an accumulation of age-related health problems [10,11,67,71,72]. Most reported physical symptoms, including pain [11,72], fatigue, and exhaustion, which have both somatic and psychic dimensions [10,11,72].

Psychosocial, spiritual, or existential factors, accounting for 21–57% of reported motives [10,11], are described as “emerging life fatigue”, “meaninglessness of life”, “tired of living”, “missing a purpose of life”, “not wanting anymore”, or “loss of dignity” [10,11,66]. This led to a reduced quality of life, deterioration of health [10,11,64,67,72], and severe suffering [60,63,66,67,69] with no hope of improvement [10,11] (“I think, she was not tired of life but simply tired of suffering” [69] p. 5). The avoidance of suffering can also refer to future and foreseeable health deterioration, as described for early dementia [60]. *Assessment of confidence in review finding no. 1: Low confidence—see*
*Table 3*.

2.Circumstances of the decision (reported by HCPs and relatives):

As reported by a Swiss study, most VSED decisions (68%) were communicated at the end of life, and less frequently at the beginning or during the course of the disease [11]. In a qualitative study from Germany, most individuals wishing to die by VSED had a terminal illness (e.g., late-stage oncological or neurodegenerative disease) with no hope for a cure [67]. In these cases, the desire underlying VSED was to end suffering by hastening foreseeable death rather than primarily to end life [67]. A study from the Netherlands reported that most patients were severely ill (76%) and that 74% had a life expectancy of less than one year [10]. However, some patients experience health deterioration without being at a terminal stage (e.g., chronic or degenerative disease), where they have little prospect of improvement [60,67]. In these cases, the desire to die persists for at least a few months [67]. The transition from symptoms and disease progression to the final decision for VSED can be fluid, with the body’s declining constitution being consciously supported [66,67,72]. However, a proportion of patients were not suffering from a severe illness (29%), according to data from a Swiss study [11], and the wish to die is sometimes motivated less by medical reasons than by old age and life fatigue [67]. One study from the Netherlands reported that 90% of VSED patients were mentally competent, 7% were only partially competent, and 2% were classified as not mentally competent [10]. Another study from Germany showed that an ethical case consultation was conducted in 27% of cases, and a psychiatric consultation in less than 10% [13,62]. *Assessment of confidence in review finding no. 2: Moderate confidence—see Table 3*.

3.Self-determination and autonomy (reported by HCPs and relatives):

The reasons for VSED are linked to a strong expression of self-determination [64,66]. Frequently reported motives relate to autonomy and independence and the fear of losing these during the disease or in the future (57–60% of motives) [10,11,66,67,72] (“… that this would at least mean a loss of independence. And for my mother […] that was something impossible: loss of independence, that was inconceivable for her.” [67] p. 6). Maintaining control over the dying process is intended [60,67,68,69], and this is seen as the last opportunity to express one’s will [66]. Accordingly, individuals opting for VSED are described as strong-minded, exhibiting a typical personality trait, as reported primarily by relatives [60,63,66,69]. Moreover, the course of VSED itself demands such will and courage (“[…] braver than asking for someone to give them a fatal injection.” [70] p. 3). *Assessment of confidence in review finding no. 3: Moderate confidence—see*
*Table 3*.

4.VSED as the “best available option” or “better than other options” (rated by relatives and HCPs):

The option of ending life through VSED was often pragmatically seen as the “best option available” or “better than other end-of-life options”. Relatives and healthcare professionals consider VSED to be preferable to the decision to commit suicide [60,68,73]. VSED was also chosen when medical aid in dying (MAiD) or death with dignity (DWD) was not available. This could be for legal reasons (not allowed in the country/state), for medical reasons (if the request was rejected because the person was not terminally ill), or if these options were not available for other reasons (e.g., cost in the U.S.) [10,60,61,63,68,71] (“She knew it was not an option, to give her medication to make her die. She wasn’t terminal by any stretch of the imagination. So I don’t think it was the ideal for her, but she was glad to have an exit strategy.” [63] p. 3). Nevertheless, cases were reported where the VSED decision was made regardless of whether access to MAiD was available [68,69]. VSED was also chosen to consciously experience the dying process, while “poisoning” through physician-assisted suicide (PAS) was not an option for one patient, as her family reported [69]. *Assessment of confidence in review finding no. 4: Moderate confidence—see*
*Table 3*.

5.Attitudes of healthcare professionals (HCPs):

Healthcare professionals seem to be open to the use of VSED. Most healthcare professionals in this analysis (>90%) accept the person’s decision for VSED and find it comprehensible [10,13,15,16,17,18,61], which is similarly the case for family physicians (94%) and nurses (92–95%) [10,15,16,18]. However, healthcare professionals also report that “food refusal” can be challenging and hard to accept [66]. According to a nationwide survey from Switzerland [15,16,17,18], most healthcare professionals perceive dying by VSED as compatible with their worldview or religion (86%) [17]. Similarly, most see it as compatible with the culture of their facility or their professional ethics (69–70%) [13,17], although physicians (58%) perceive it as less compatible than nurses (75–80%) [15,16,18]. Despite an overall positive view on VSED, the responses from the studies on whether physicians would recommend VSED vary between the Netherlands (34%) [10] on the one hand, and Switzerland and Germany (58–61%) on the other [15,62]. Nurses seem to be even more reluctant in this regard [73]. Less than half of nurses (33–48%) would recommend VSED to a patient [16,18]. 

Although the majority (77%) of healthcare professionals agree with clarifying the patient’s competence, 13% have a neutral position, and 10% believe that judging the patient’s competence is unnecessary, as reported by the nationwide Swiss study [17]. This judgement was significantly more critical for healthcare professionals without VSED experience than those with experience [17]. Finally, attitudes may also differ according to cultural background, as seen in studies from Switzerland, where some attitudes differ between the various linguistic and cultural regions. For example, in the Lake Geneva region, VSED is less likely to be considered a dignified death than in other regions [17]. *Assessment of confidence in review finding no. 5: Low confidence—see Table 3*.

6.Attitudes towards VSED as natural dying or suicide (rated by HCPs):

The attitudes towards VSED range from “dying naturally” to “letting die” or “something else” to “suicide”. Most healthcare professionals (63%) [17] consider death by VSED to be comparable to a natural death [15,16,18,66,68,69,73] or would agree that supporting a patient during VSED does not constitute assisted suicide (56%) [13]. More nurses (64–70%) [16,18] than physicians (59%) [15] consider it a natural death in Switzerland. Moreover, most VSED cases (81–99%) are documented as “natural death” on the death certificate [11,71]. However, this fact should be interpreted cautiously, as there may be other reasons (e.g., medico-legal) besides the physicians’ attitude. In addition to healthcare professionals, relatives had not rated their loved one’s VSED as suicide, as reported in a qualitative study from Switzerland [69]. About one-quarter (26.5%) of healthcare professionals view VSED as equivalent to “letting die”, and some see it as “something else” (approx. 5%) [15,16,17,18,73]. In some cases (approx. 5%), VSED is regarded as (medically assisted) suicide [15,16,17,18,66,73]. The assessment of VSED as “natural dying” or “suicide” can also depend on the specific cases and their circumstances (e.g., the age, disease, and life expectancy of the patient) [13,65,69]. *Assessment of confidence in review finding no. 6: Very low confidence—see Table 3*.

7.Acceptance and its influencing factors (rated by HCPs and relatives):

The acceptance of a VSED decision depends on several factors. These are patient characteristics, the assessment of whether VSED is a suicide, the circumstances surrounding the decision, and personal characteristics.

Patient characteristics: Acceptance of the VSED decision is primarily related to the person’s state of health [13,60,63,64,69,70,73]. The more visibly ill a person is and the more apparent the current symptom burden, the more understandable and acceptable the decision for VSED is. The situation of terminally ill patients who “hasten it a little” [60] was primarily understood and accepted [13,60,70]. When persons are in good health or not visibly ill, for instance, when deterioration is expected in the future (e.g., early dementia) [13,63], the decision for VSED is less accepted [13,63,64,69,73] (“The healthier the patient, the more difficult” [73] p. 8). The patient’s age may also influence whether their decision for VSED is accepted. A younger person’s decision to use VSED tends to be less understandable and acceptable to others [60,73] than older individuals who choose VSED [69]. 

The assessment of whether VSED is suicide: Healthcare professionals, relatives, and facilities who regard VSED as suicide often reject VSED [60,66,68,69], whereas people who regard VSED as natural dying are more likely to accept the VSED decision [69,70]. However, the data also show that physicians classifying VSED as suicide are open to VSED and would be willing to accompany a patient [15]. 

Circumstances surrounding the decision: For the acceptance of the VSED decision, the traceability of the motives underlying this decision is essential [60,66,68,73]. A prerequisite for understanding and accepting the VSED decision is to get in touch with the patients and know their personalities and situations very well [60,66,69,73]. Moreover, reasons related to self-determination and independence are generally well-accepted by healthcare professionals [17,61,66,69,70] and relatives [60,64].

Personal characteristics: Acceptance can also be influenced by culture, religion, and faith, as well as experience and knowledge of VSED [60,64,69,73]. For instance, there are cases reported in which the cultural background or religious convictions stood in the way of accepting the VSED decision [60,64,73]. However, studies have not found any significant differences in the degree of religiosity [13] or between different denominations [17]. Furthermore, knowledge about VSED and previous VSED experiences are associated with higher acceptance of the decision [15,17,69], and a lack of knowledge, for example, about the legal situation, can make acceptance more difficult [73]. *Assessment of confidence in review finding no. 7: Low confidence—see Table 3*.

8.Advocacy by healthcare professionals (HCPs):

According to a nationwide Swiss study, a healthcare professional accompanies most VSED cases: 79% by nurses and 72% by family physicians [11]. This Swiss data revealed that the willingness to accompany a person during VSED could be high (92–95%) [15,16,17,18]. If the nurses had been caring for the patient for years and the decision was in line with the patient’s personality and the course of the illness, it was obvious for them to accompany the person [69]. Once the decision had been made to accompany them, healthcare professionals could feel obliged to support the patient’s wish to die through VSED and can evolve into advocates for the person who wishes to die. They are committed to the patient’s needs and do everything they can to provide the person with the support and environment they need [60,69,73] (“That is my task just as much as supporting or helping him or offering him services. That I also stand up for him and stand up for him and say: that is his wish (…) actually almost as an advocate.” [73] p. 8). Advocacy may go so far that healthcare professionals act independently of the team or even feel compelled to act against the facility’s decision (in cases of interdiction of VSED accompaniment) to ensure the successful fulfillment of the patient’s wish [68,69]. The patient’s “happiness” and trust were described as fulfilling, making care and engagement even easier [68,69]. *Assessment of confidence in review finding no. 8: Moderate confidence—see*
*Table 3*.

9.Advocacy by relatives:

Just over half (58%) of VSED cases are accompanied by relatives, as reported by the nationwide Swiss study [11]. Once they have accepted the wish to die by VSED, relatives usually agree to accompany their loved ones [60,64,69]. Even in cases of rejection or lack of understanding, relatives believe that people have the right to choose their way of dying [63,64]. Once involved, relatives often play an active role and act as the patient’s representative (“I promised her that I would fight to the death. Fight to her will!” [64] p. 7). They speak for and on behalf of the patient, take the lead in contacting physicians or hospices, and make logistical arrangements to achieve successful VSED [60,63,64]. They coordinate and organize farewells to family and friends and do everything essential for the patient [60,63]. They try to reduce contact with tempting food and drink [63] or provide practically around-the-clock monitoring to ensure that patients are not given drinks or food against their will or forced to eat and drink [64]. They do all this for as long as it is necessary and despite their insecurities, grief, or other obstacles, such as rejection and accusations from others [60,64] (“And I just had to look at their faces. They clearly showed: ‘Do you want to kill your mother?’ And that’s very depressing.” [64] p. 5). These obstacles and challenges proved conducive to becoming the patient’s advocate [64]. This role becomes even more critical as the patient’s capacity declines, especially after losing consciousness. At this point, relatives take full responsibility for the successful course of VSED [60,63,64]. *Assessment of confidence in review finding no. 9: Moderate confidence—see*
*Table 3*.

10.The grieving process after VSED accompaniment:

In the case of a terminal illness, grief after VSED is not perceived as more complicated, and the loss of the deceased person was more important to the grief than VSED itself [67]. Compared to grief after a suicide, the situation was rated as “better” [67]. This was linked to the opportunity to say goodbye and communicate with the loved one and the possibility of professional support [67]. The tireless support and advocacy during VSED can lead to excessive exhaustion for the relatives. While grieving, they try to “recharge their batteries”, reflect on what has happened, and take the time to mourn consciously [64].

Nevertheless, how relatives perceive the grieving period depends on how they assess their accompanying and the dying situation. They can feel fulfilled, satisfied, and grateful if they rate both as positive [64] (“These were the most beautiful fifteen days in my life with my mother. Yes. Because I could do anything for her.” [64] p. 8). However, if the dying process was not perceived as peaceful, for example, if the person was suffering or the death was agonizing, this made the grieving process more difficult [64,67] (“I just keep seeing this picture from the last three days. It just wasn’t good. It’s not a good way to die […] If you ask me, I say perish” [64] p. 8). When relatives experience a situation that is not manageable due to a lack of support or when they struggle with their support and advocacy, they have to fight feelings of guilt and self-blame, which can prolong grief [64,67]. Grief was also experienced as challenging and burdensome when the wish to hasten death was not understandable, or the decision for VSED itself was doubted, for example, when relatives rejected the VSED as suicide [67]. These negative experiences and evaluations can lead to traumatic memories that cause a high level of distress [64,70]. *Assessment of confidence in review finding no. 10: Moderate confidence—see*
*Table 3*.

### 3.2. Assessing Confidence in the Review Findings and Geographical Limitations

Assessing confidence in the review findings:

The quality assessments of nineteen publications [10,11,12,13,14,15,16,17,18,60,61,63,64,65,66,67,68,69,70,71,73] have already been published [48]. The quality assessment of three publications [11,71,72] can be found in the Appendix A). Overall, the quality of the qualitative research was rated as high. One study, whose quality assessment was published here, was rated as low quality [11]. This was mainly owing to the sampling strategy for VSED cases, which led to a high risk of recall bias and thus raised concerns regarding the quantification of the statements in this study. The contribution of this study to individual review findings and its contribution to confidence in those were assessed as part of the CERQual assessment (Appendix A). In this CERQual assessment, six review findings were rated with moderate confidence (Finding nos. 2, 3, 4, 8, 9, and 10), three with low confidence (Finding nos. 1, 5, and 7), and one with very low confidence (Finding no. 6) (Table 3). For the first review finding “the motives—high symptom burden and suffering”, there were concerns regarding indirect evidence as only studies from the perspective of healthcare professionals and relatives, and not from patients directly, were available at the time of data synthesis.

2.Geographical limitations:

The main concerns relate to the component of “relevance”. There were no geographical restrictions for the systematic search and, thus, for the research question. However, studies from only six countries were identified (Table 4), primarily from Switzerland (50%). Most identified studies (75%) are from Europe, increasing to 100% when focusing on only quantitative studies. This raises serious concerns about generalizing the results to a global context (partial relevance), and the results (nos. 1–10) must be interpreted cautiously. Accordingly, this was the main reason for downgrading the confidence in the CERQual approach (Table 3).

## 4. Discussion

### 4.1. Main Findings

The results focused on three main themes. First, the motives and situation of decision-making; second, attitudes towards VSED, primarily from the perspective of healthcare professionals; and third, the advocacy that occurs with close VSED support, including the grief of relatives following the death of a loved one through VSED.

People who opt for VSED usually have a high symptom burden and suffer from severe or terminal illnesses. However, patients who anticipate that their health will deteriorate in the future also decide to undergo VSED. The stated motives for VSED are often related to expressing self-determination and autonomy, and show the fear of losing these. The decision for VSED is frequently described pragmatically as the “best available option” or “better than other options”. Most healthcare professionals accept people’s decisions regarding VSED and find them understandable. The majority of healthcare professionals see VSED as natural dying, some as “letting die”, and a minority as suicide. Acceptance of a VSED decision depends on several factors, primarily the person’s health. The more apparent the symptom burden and the more visible the illness, the better the decision is accepted by the environment. Generally, classifying a VSED as natural dying leads to greater acceptance than categorizing it as suicide. Personal attitudes, cultural backgrounds, and religious beliefs sometimes make acceptance difficult. However, knowledge of VSED and previous VSED experiences are associated with a higher acceptance of VSED decisions.

The willingness to accompany another person during VSED was high among healthcare professionals. If they are committed to the patient’s wish for VSED, they can evolve into advocates for the dying person by doing whatever is necessary to provide the best possible support. Advocacy can go so far that healthcare professionals feel compelled to act secretly against institutional decisions. Relatives often play an active role in VSED by managing everything necessary and protecting the patient’s will and wishes against obstacles and opposition from those around them. This tireless commitment can lead to excessive exhaustion, and after the death of a loved one, relatives have to “recharge their batteries”. Depending on how they assess their support and the dying situation, relatives can feel fulfilled and grateful but may also experience a more difficult mourning phase.

### 4.2. Voluntary Stopping of Eating and Drinking as “Natural Death” or “Suicide”

The attitudes of healthcare professionals towards VSED as a natural death or suicide vary (Finding no. 6). Over half of the surveyed professionals consider VSED a natural death, while about one-quarter view VSED as equivalent to “letting die”, which is closest to the term of “passive euthanasia” [17]. A minority categorize it as suicide. Since quantitative data derive from one Swiss and one German survey, final conclusions about quantifications must be drawn carefully. However, qualitative data from Switzerland, Germany, and the U.S. support different aspects of this finding. Notably, these attitudes towards VSED as natural death or suicide reflect the healthcare professionals’ perceptions and must be distinguished from the legal and normative classifications. Nevertheless, empirical data on healthcare professionals’ perspectives are invaluable for informing and supporting this ethical and legal debate.

Factors that may influence these views include the patient’s age, and their life expectancy is probably the most decisive factor (finding No. 6). The study by Stängle et al. indicates that participants with VSED experience are more likely to see VSED as a natural death and less often as suicide compared to participants without VSED experience [17]. Participants in a U.S. study found it challenging to distinguish between deaths by suicide and deaths that were intentionally hastened but not considered suicide, noting that the boundaries between these categories are blurred [68]. Participants from the same study described “suicide” as a loaded term, associated with religious connotations and painful emotions and meanings, regarding it as a “huge barrier” [68].

The assessment of whether VSED is considered suicide has practical implications for healthcare professionals. Viewing VSED as suicide can lead to its rejection by healthcare professionals and institutions [60,66,68,69]. Both may experience discomfort when deciding on VSED accompaniment. Depending on the country’s legislation, fear and uncertainties may arise concerning the legal situation and whether the accompaniment of VSED is legal or has to be regarded as a form of medically assisted suicide [68,69,70]. Ideally, there is a consistent attitude within institutions and teams regarding VSED and its assessment as natural death or suicide. Discrepant attitudes can create a dilemma for healthcare professionals between personal values and professional obligations. There are examples of institutions prohibiting accompaniment during VSED because they consider VSED to be suicide. In these cases, healthcare professionals who felt committed to patients’ wishes found themselves compelled to act contrary to the institution’s decision [68,69]. Moreover, physicians are confronted with difficulties in correctly completing the death certificate, as they must decide whether the death should be rated as “natural” or “unnatural”. Choosing “unnatural” may have problematic legal or regulatory consequences [22,28,29,74]. To date, there is no consensus among experts regarding the classification of VSED. Nevertheless, guidance is necessary to support physicians in addressing this issue, and solutions have been proposed [28,74].

### 4.3. What This Paper Contributes and Comparisons with Existing Evidence Synthesis

We present a range of attitudes of healthcare professionals towards VSED based on empirical data, providing a broad and nuanced view of existing opinions. Even though the data analyzed here reveal that healthcare professionals have a positive attitude towards VSED, it is essential to reflect the entire spectrum of opinions and to capture attitudes that differ from the majority to fully reflect the complexity of VSED. Despite the acceptance of VSED, we identified factors that influence acceptance, highlighting how it depends on the individuals involved and their unique situations. We show that relatives and healthcare professionals can become natural advocates, emphasizing that accompanying VSED can be a challenging and extraordinary process in which healthcare professionals can feel deeply involved. Although the data show that grief following VSED is generally not different from grief following death without VSED, in certain circumstances, there is a risk that the grieving process may be more difficult. This highlights the importance of relatives being supported during VSED and after the death of their loved one.

Furthermore, our results confirm the findings of the first evidence synthesis from 2014 [1]: patients’ motives, such as the deterioration of health, poor quality of life, fatigue with life, and motives related to self-determination and autonomy (Finding nos. 1 and 3). Additionally, the approval rate of healthcare professionals for accompanying a person during VSED was confirmed (Finding no. 8). However, the results from 2014 were based on a small number of quantitative studies, and the authors stated that the results should be interpreted cautiously. Our review findings were supported by a combination of quantitative and qualitative data, both of which strengthened the scientific rigor of the results and provided a differentiated view of VSED. Here, we included the perspectives of physicians and relatives, and thus, a perspective close to the patient.

An integrated review focusing only on the experiences of relatives based on four studies, all of which were also included in this systematic review, reached similar conclusions about the accompaniment and advocacy of relatives [75]. In addition to the authors’ first publication on the initial set of results [48], a further systematic review was published at the end of 2024 [76], reflecting the ongoing scientific interest in this important topic.

### 4.4. Confidence Assessments with CERQual and Geographical Limitations

In general, the assessment of the findings on “attitudes towards VSED” (Finding nos. 5 and 6) gives more cause for concern than the findings dealing with caring, human solidarity, family cohesion, or grief as fundamental human issues (Finding nos. 8, 9, and 10). Attitudes are assumed to be more dependent on culture and religious background than fundamental human issues. Stängle et al. revealed that attitudes towards VSED vary among different linguistic and cultural regions within Switzerland. Other empirical data indicate a link between attitudes and faith [60,64,69,73]. It is likely that attitudes towards VSED differ between countries where religion plays a major role and those that are more secular. In countries where talking openly about dying or end-of-life options is considered taboo, VSED may be viewed more critically.

The majority of identified studies (75%) originate from Europe, specifically Switzerland (four studies), the Netherlands (two studies), and Germany (two studies). Two studies originated from the U.S. and one from New Zealand. In these countries and in nine U.S. states, medical aid in dying is legal or not explicitly prohibited [77]. Notably, euthanasia is legal in the Netherlands and New Zealand. Furthermore, in the Netherlands and Switzerland, right-to-die organizations have a long tradition, and end-of-life options are discussed more openly in public discourse than in other countries. These circumstances are likely to influence the attitudes of the study participants and may contribute to an overall trend towards more positive opinions. Therefore, we presume that future studies from countries with a more critical perspective could alter the findings on “attitudes towards VSED”. Notably, in addition to positive findings, burdensome and challenging experiences of healthcare professionals and relatives were also reported from these countries [48].

### 4.5. Implications for Research

Since our search in November 2021, five empirical studies have been published, showing that research in the field of VSED is continuing and growing [78,79,80,81,82]. These studies were conducted in Switzerland [78], the U.S. [79], the Netherlands [80], and Germany [81,82] and included different perspectives (nurses, counsellors, ethicists, medical doctors, politicians, volunteers, relatives, the religious community, patients, professional caregivers, the palliative care team, and medical students) and both qualitative [78,79,80] and quantitative study designs [81,82]. The study by Bolt and colleagues is particularly noteworthy as it is the first patient-oriented study to provide direct insights into patient-relevant issues and is also relevant to the patients themselves, as their situation, characteristics, and needs were directly examined [80]. As the studies were mainly carried out in the same countries, studies from other countries would be valuable in further completing the VSED picture.

### 4.6. Strengths and Limitations

This systematic review provides an exhaustive summary of the current evidence from primary studies on the attitudes and experiences of people confronted with VSED. Different perspectives were captured from healthcare professionals and relatives. This systematic review included rigorous data synthesis and analysis methods. Quantitative studies supported the data quantity, while qualitative studies supported data richness. Additionally, this review strengthens the reliability of the presented results by assessing the confidence of review findings using the GRADE-CERQual approach, a structured and transparent method that plays a vital role in informing people during decision-making, guideline development, and policy [52].

As outlined in Section 4.5, five additional empirical studies [78,79,80,81,82] have been published since our systematic search in 2021. This constitutes a limitation for the results of our analysis. Another limitation is the language restrictions defined for resource reasons. Studies on this topic may be published in local languages and possibly not in international journals or databases.

There is a potential risk that participants with a more positive view of VSED could have been overrepresented in the selection and recruitment of study participants. Some study authors have also addressed these concerns.

## 5. Conclusions

VSED is a multi-layered phenomenon that affects patients and all caregivers, such as nurses, physicians, and relatives who are confronted with it. We provide an empirical and evidence-based foundation for scientific, ethical, and legal discussions, debates, and controversies regarding VSED by synthesizing real-world data on the attitudes and experiences of people who directly encounter it. Furthermore, by assessing the confidence of these findings, we provide a more nuanced view of the highly complex phenomenon of VSED.

## Figures and Tables

**Table 1 healthcare-13-01264-t001:** Overview of the included studies (*n* = 16).

Study Design	Perspective	Country	Reference
Qualitative studies (*n* = 7)	Relatives	The United States	Lowers et al., 2021 [63] and Lowers 2020 [60]
	Relatives	Switzerland	Eppel-Meichlinger et al., 2021 [64]; Stängle et al., 2019a [65]
	HCPs	Switzerland	Fringer et al., 2020 [66]
	Relatives	Germany	Starke 2020 [67]
	HCPs	The United States	Gerson et al., 2020 [68] and Gerson 2018 [61]
	HCPs and Relatives	Switzerland	Saladin et al., 2018 [69]
	Physicians	New Zealand	Malpas & Mitchell 2017 [70]
Quantitative studies (*n* = 8)	Physicians	The Netherlands	Hagens et al., 2021 [71]
	Nurses (heads of outpatient care)	Switzerland	Stängle et al., 2021c [18]
	HCPs	Switzerland	Stängle et al., 2021b [17]; Stängle et al., 2019b [72]; Stängle et al., 2019c [11]
	Physicians	Switzerland	Stängle et al., 2020a [15]
	Nurses (heads of long-term care)	Switzerland	Stängle et al., 2020b [16]
	Physicians	Japan	Shinjo et al., 2019 [14] ^2^
	Physicians	The Netherlands	Bolt et al., 2015 [10]
	Physicians	Germany	Hoekstra et al., 2015 [13] and Hoekstra 2020 [62]
Mixed-methods study (*n* = 1)	HCPs and divers ^1^	Switzerland	Stängle and Fringer 2021a [73]

HCPs: Healthcare professionals (mainly physicians and nurses); ^1^ counsellors, ethicists, politicians, volunteers, and relatives (with only one counsellor, one politician, and one volunteer participating in the study, as noted in [73]). ^2^ The data from this survey did not contribute to the results published here. Nevertheless, it is listed here for completeness, as it was identified by the systematic search.

**Table 2 healthcare-13-01264-t002:** Overview of review findings.

**A: Motives and Decision-making**
1. Motives—high symptom burden and suffering
2. Circumstances of the VSED decision
3. Self-determination and autonomy
4. VSED as the “best available option” or “better than other options”
**B: Attitudes towards VSED from healthcare professionals and relatives**
5. Attitudes of healthcare professionals
6. Attitudes towards VSED as natural dying or suicide
7. Acceptance and its influencing factors
**C: Advocacy during accompaniment and relatives’ grief**
8. Advocacy by healthcare professionals
9. Advocacy by relatives
10. Grieving process after VSED accompaniment

The main themes (A, B, and C) and subthemes (1.–10.), which are the review findings, are presented here.

**Table 3 healthcare-13-01264-t003:** Summary of the GRADE-CERQual assessment for the ten review findings.

**#**	**Summarized Review Finding**	**CERQual ** **Assessment of ** **Confidence in the Evidence**	**Explanation of ** **CERQual Assessment**	**Studies Contributing to the Review Finding**
**A: MOTIVES AND DECISION-MAKING**				
1	Motives—high symptom burden and suffering: Individuals opting for VSED typically have a high symptom burden and an accumulation of health problems, which finally result in a reduced quality of life and the deterioration of health status. Psychosocial, spiritual, or existential factors such as “emerging life fatigue” or “meaninglessness of life” also play a role in the decision-making process for VSED. Avoidance of suffering can also refer to the future, as described for early dementia.	**Low** **confidence**	Minor concerns regarding methodological limitations. Minor concerns regarding coherence. No/very minor concerns regarding adequacy. There are serious concerns regarding relevance because of indirect evidence (findings on the motives of VSED persons, but data from attending physicians, nurses, and relatives) and partial relevance (global research question, but the review finding is based only on data from three countries, primarily from Europe).	Bolt et al., 2015 [10]; Eppel-Meichlinger et al., 2021 [64]; Fringer et al., 2020 [66]; Hagens et al., 2021 [71]; Lowers 2020 [60]; Lowers et al., 2021 [64]; Saladin et al., 2018 [69]; Stängle et al., 2019b [72]; Stängle et al., 2019c [11]; Starke 2020 [67]
2	Circumstances of the VSED decision: Persons who wished to die by VSED often had a terminal illness. Most patients had severe disease (76%) and a life expectancy of less than one year (74%). The transition from symptoms and disease progression to the final decision of VSED can be fluid, with conscious support for the body’s declining constitution. However, patients also experience deterioration in their health without being at a terminal stage, and one-third of patients did not suffer from a severe illness (29%). Not all VSED patients (90%) were assessed as competent to opt for VSED. Ethical case consultations were conducted in 27% of cases, and a psychiatric consultation in less than 10% of VSED cases.	**Moderate confidence**	Minor concerns regarding methodological limitations. No/very minor concerns regarding coherence. There are moderate concerns regarding adequacy (missing data quantity for the individual aspects of the review finding). There are serious concerns regarding relevance (global research question, but the review finding is based only on data from four countries, primarily from Europe).	Bolt et al., 2015 [10]; Fringer et al., 2020 [66]; Hoekstra et al., 2015 [13]; Hoekstra 2020 [62]; Lowers 2020 [60]; Stängle et al., 2019b [72]; Stängle et al., 2019c [11]; Starke 2020 [67]
3	Self-determination and autonomy: Frequently reported motives for VSED are often associated with autonomy and independence and the fear of losing them during the disease (57–60% of the motives). Retaining control over the dying process is desired; the last resort is expressing one’s will. Accordingly, persons who have chosen VSED are described as strong-minded, a characteristic that the course of VSED itself requires.	**Moderate confidence**	Minor concerns regarding methodological limitations. No/very minor concerns regarding coherence. No/very minor concerns regarding adequacy. There are moderate concerns regarding relevance (global research question, but the review finding is based only on data from five countries).	Bolt et al., 2015 [10]; Eppel-Meichlinger et al., 2021 [64]; Fringer et al., 2020 [66]; Gerson et al., 2020 [68]; Lowers 2020 [60]; Lowers et al., 2021 [64]; Malpas & Mitchell 2017 [70]; Saladin et al., 2018 [69]; Stängle et al., 2019b [72]; Stängle et al., 2019c [11]; Starke 2020 [67]
4	VSED as the “best available option” or “better than other options”: The option of ending life through VSED was often pragmatically regarded as the “best option available” or “better than other end-of-life options”. Compared to the decision to commit suicide, VSED was preferred. The same was true if it was the only option when physician-assisted suicide or medically assisted suicide was not available (e.g., because of jurisdiction, the lack of a terminal condition, or costs). Additionally, VSED was also chosen in order to experience the dying process consciously.	**Moderate confidence**	No/very minor concerns regarding methodological limitations. No/very minor concerns regarding coherence. There are minor concerns regarding adequacy (could be more quantitative data). There are serious concerns regarding relevance (global research question, but the review finding is based only on data from three countries).	Bolt et al., 2015 [10]; Gerson 2018 [61]; Gerson et al., 2020 [68]; Hagens et al., 2021 [71]; Lowers 2020 [60]; Lowers et al., 2021 [64]; Saladin et al., 2018 [69]; Stängle & Fringer 2021a [73]
**B: ATTITUDES TO VSED FROM HEALTHCARE PROFESSIONALS AND RELATIVES**				
5	Attitudes of healthcare professionals (HCPs): HCPs seem to be open to VSED. They (>90%) largely accept the person’s decision for VSED and perceive it as understandable, although they also report that “food refusal” can be a challenging topic. Most HCPs consider dying by VSED as compatible with their worldview or religion (86%) and as compatible with the culture of their facility or their professional ethics (69–70%). Despite these positive attitudes, physicians (34–61%) and nurses (33–48%) are more reluctant to recommend VSED. Although the majority (77%) of HCPs believe it is important to judge the patients’ mental capacity when they opt for VSED, a fifth (22%) take a neutral position or believe that this is unnecessary. This judgment was significantly more critical for HCPs without VSED experience than for those with experience. Furthermore, attitudes may differ across language and cultural regions.	**Low** **confidence**	No/very minor concerns regarding methodological limitations. Minor concerns regarding coherence. There are moderate concerns regarding adequacy (mainly quantitative data; some aspects based primarily on one nationwide study). There are serious concerns regarding relevance (it is not unlikely that the results regarding attitudes to this controversial topic will change if studies from other and more critical countries are included).	Bolt et al., 2015 [10]; Fringer et al., 2020 [66]; Gerson 2018 [61]; Hoekstra et al., 2015 [13]; Hoekstra 2020 [62]; Stängle et al., 2020a [15]; Stängle et al., 2020b [16]; Stängle & Fringer 2021a [73]; Stängle et al., 2021b [17]; Stängle et al., 2021c [18]
6	Attitudes towards VSED as natural dying or suicide: Attitudes on VSED vary. Most healthcare professionals (63%) view VSED as a natural death, with more nurses (67%) than physicians (59%) considering it that way. In addition, VSED is equated with “letting die” by a proportion of professionals (27%), and a minority regard it as “something else” (about 5%). A minority (about 5%) see VSED as (assisted) suicide. Most VSED cases (81–99%) are documented as “natural death” on the death certificate. In addition to healthcare professionals, it was also reported that relatives did not classify their loved one’s VSED as an act of self-killing. The assessment of VSED may also depend on the specific case and circumstances (e.g., the age, disease, and life expectancy of the VSED patient).	**Very low confidence**	Minor concerns regarding methodological limitations. There are moderate concerns regarding coherence (contradictory data and alternative explanations with two individual aspects). There are moderate concerns regarding adequacy (data quantity and richness could be improved). There are serious concerns regarding relevance (it is not unlikely that the results regarding attitudes to this controversial topic will change if studies from other and/or more critical countries are included).	Fringer et al., 2020 [66]; Gerson et al., 2020 [68]; Hagens et al., 2021 [71]; Hoekstra et al., 2015 [13]; Saladin et al., 2018 [69]; Stängle et al., 2019c [11]; Stängle et al., 2020a [15]; Stängle et al., 2020b [16]; Stängle & Fringer 2021a [73]; Stängle et al., 2021b [17]; Stängle et al., 2021c [18]; Starke 2020 [67]
7	Acceptance and its influencing factors: The acceptance of a VSED decision by healthcare professionals (HCPs) and relatives depends on several factors: (a) Patient characteristics: Acceptance is primarily related to the person’s health status. For example, in the situation of a terminally ill patient, VSED is widely accepted. In contrast, in the case of a person in good health or without visible disease, it is less accepted. Acceptance also appears to increase with the age of the person concerned. (b) Assessment of whether VSED is suicide: The assessment of the situation as suicide by HCPs, relatives, and facilities may lead to their rejection of VSED, whereas the definition of natural dying generally leads to more acceptance. (c) Circumstances surrounding the decision: Traceability of motives and knowing the patient and their situation well were critical to promote acceptance. Furthermore, reasons related to self-determination and independence were generally well accepted by HCPs and relatives. (d) Personal characteristics: Sometimes, personal attitudes, cultural backgrounds, or religious beliefs do not allow the acceptance of VSED.	**Low** **confidence**	No/very minor concerns regarding methodological limitations. No/very minor concerns regarding coherence. There are moderate concerns regarding adequacy. There are serious concerns regarding relevance (global research question, but the review finding is only based on data from four countries, especially when looking at the individual aspects of the review findings).	Eppel-Meichlinger et al., 2021 [64]; Fringer et al., 2020 [66]; Gerson 2018 [61]; Gerson et al., 2020 [68]; Hoekstra et al., 2015 [13]; Lowers 2020 [60]; Lowers et al., 2021 [64]; Malpas & Mitchell 2017 [70]; Saladin et al., 2018 [69]; Stängle et al., 2020a [15]; Stängle & Fringer 2021a [73]; Stängle et al., 2021b [17]
**C: ADVOCACY DURING ACCOMPANIMENT AND RELATIVES’ GRIEF**				
8	Advocacy by healthcare professionals (HCPs): Most VSED cases (76%) are accompanied by HCPs, and their willingness to accompany a person during VSED is high (>90%). Once committed to the patient’s wish, they can evolve into an advocate for the VSED person. When caregiving has been ongoing for years, the decision to provide accompaniment is evident to them. They are committed to the patient’s needs and wishes and do everything they can to support the patients. Advocacy may be to the point where HCPs act independently of the team or feel compelled to act contrary to the facility’s decision (in cases of interdiction of VSED accompaniment). The patient’s “happiness” and trust make care and engagement even more accessible, as they reported.	**Moderate confidence**	Minor concerns regarding methodological limitations. No/very minor concerns regarding coherence. Minor concerns regarding adequacy. There are serious concerns regarding relevance (global research question, but the review finding is based only on data from two countries, primarily from Switzerland). As the content of the review finding deals with caring, professional self-conception, and human solidarity, and is less about attitudes, the differences between the countries were judged to not be so great (leading to the downgrade to low confidence and not to very low confidence).	Gerson et al., 2020 [68]; Lowers 2020 [60]; Saladin et al., 2018 [69]; Stängle et al., 2019c [11]; Stängle et al., 2020a [15]; Stängle et al., 2020b [16]; Stängle & Fringer 2021a [73]; Stängle et al., 2021b [17]; Stängle et al., 2021c [18]
9	Advocacy by relatives: Just over half (58%) of VSED cases are accompanied by relatives. They often play an active role, doing whatever is necessary for the patient: speaking for and on behalf of the patient, making logistical arrangements, organizing farewells to family and friends, protecting the patient from exposure to tempting food and drink and the forced administration of the same. They discuss and debate with professionals or other family members to ensure that their loved one’s will is respected. They do all this despite their own grief and resistance, such as legal or moral accusations. These obstacles have even led them to become advocates. With diminishing capacity and unconsciousness, the role of advocacy is even more important to ensure the success of VSED.	**Moderate confidence**	Minor concerns regarding methodological limitations. No/very minor concerns regarding coherence. There are moderate concerns regarding adequacy (individual aspects should be supported by further research). There are serious concerns regarding relevance (global research question, but the review finding is based only on data from two countries). As the content is more about human solidarity and family cohesion and less about attitudes, the differences between the countries were judged to not be so great (leading to downgrading to low confidence and not to very low confidence).	Eppel-Meichlinger et al., 2021 [64]; Lowers 2020 [60]; Lowers et al., 2021 [64]; Saladin et al., 2018 [69]; Stängle et al., 2019c [11]
10	Grieving process after VSED accompaniment: In the case of a terminal illness, grief after a VSED is not perceived as more complicated than without VSED. Compared to suicide, it is rated as “better” because of the opportunity to say goodbye and seek professional support. Relatives provide tireless support, which leads to excessive exhaustion. During the grieving phase, they “recharge their batteries” and reflect on what has happened. Relatives can feel fulfilled, satisfied, and grateful, depending on how they assess their accompaniment and the dying situation. However, they may experience a more difficult grief, for example, when the VSED person had suffered, when the death was agonizing, when they have experienced a situation that was not manageable due to a lack of support, or when they are struggling with their support. If the wish to hasten death was not understandable, or if they reject VSED as suicide, they have to fight feelings of guilt and self-blame. These negative experiences and assessments can lead to traumatic memories and distress.	**Moderate confidence**	No/very minor concerns regarding methodological limitations. No/very minor concerns regarding coherence. There are moderate concerns regarding adequacy (review content should be supported by further research). There are serious concerns regarding relevance (global research question, but the review finding is based only on data from three countries). As the content is more about human solidarity and family cohesion and less about attitudes, the differences between the countries were judged to not be so great (leading to downgrading to low confidence and not to very low confidence).	Eppel-Meichlinger et al., 2021 [64]; Malpas & Mitchell 2017 [70]; Starke 2020 [67]

VSED, voluntary stopping of eating and drinking; HCPs, healthcare professionals.

**Table 4 healthcare-13-01264-t004:** Distribution of studies by country of origin.

Origin of Countries (*n* = 6)	Number of Studies (*n* = 16/100%)	Number of Qualitative Analyses	Number of Quantitative Analyses (%)/(Number of Participants)	Number of Mixed-Methods Analyses
Switzerland	8 (50%)	3 (18.75%)	4 (25%) ^1^ (*n* = 1681)	1 (6.25%) ^1^
The Netherlands	2 (12.5%)	0 (0%)	2 (12.5%) (*n* = 733)	0 (0%)
Germany	2 (12.5%)	1 (6.25%)	1 (6.25%) (*n* = 255)	0 (0%)
The United States	2 (12.5%)	2 (12.5%)	0 (0%)	0 (0%)
New Zealand	1 (6.25%)	1 (6.25%)	0 (0%)	0 (0%)
Japan	1 (6.25%)	0 (0%)	1 (6.25%) (*n* = 571) ^2^	0 (0%)

^1^ The quantitative data (including the quantitative parts of mixed-methods analyses) derive from one nationwide survey with 1681 participants (751 family physicians, 535 heads of long-term care, and 395 heads of outpatient care) and are published in five articles. ^2^ The data from this survey did not contribute to the results published here. Nevertheless, it is listed here for completeness, as it was identified by the systematic search.

## Data Availability

A more detailed description of the methods and the individual steps of the systematic search, including search strategy development, data analysis, quality assessment, and confidence assessment, can be found in [48] and its supplementary materials, and in the Appendix A of this publication. Furthermore, all Excel files documenting the search, abstract screening, and full-text screening, and the MAXQDA file (MAXQDA 2022, version 22.8.0) with all extracted data and codes, are available from the corresponding author upon reasonable request.

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
