# Peer review of "Carers’ Perspective on Voluntary Stopping of Eating and Drinking: A Systematic Mixed-Methods Review of Motives and Attitudes"

_healthcare, 2025, doi:10.3390/healthcare13111264_

Round 1
Reviewer 1 Report
Comments and Suggestions for Authors
See attached

Author Response
Response to Reviewer 1 Comments
|
||||||||||||||||
1. Summary |
|
|
||||||||||||||
Thank you very much for taking the time to review this manuscript. Please find the detailed responses below and the corresponding revisions/corrections highlighted/in track changes in the re-submitted files. |
||||||||||||||||
2. Questions for General Evaluation |
Reviewer’s Evaluation |
Response and Revisions |
||||||||||||||
Does the introduction provide sufficient background and include all relevant references? |
Yes/Can be improved/Must be improved/Not applicable |
|
||||||||||||||
Are all the cited references relevant to the research? |
Yes/Can be improved/Must be improved/Not applicable |
|
||||||||||||||
Is the research design appropriate? |
Yes/Can be improved/Must be improved/Not applicable |
Please find our answers below |
||||||||||||||
Are the methods adequately described? |
Yes/Can be improved/Must be improved/Not applicable |
|
||||||||||||||
Are the results clearly presented? |
Yes/Can be improved/Must be improved/Not applicable |
Please find our answers below |
||||||||||||||
Are the conclusions supported by the results? |
Yes/Can be improved/Must be improved/Not applicable |
Please find our answers below |
||||||||||||||
3. Point-by-point response to Comments and Suggestions for Authors |
||||||||||||||||
Comments 1: Geographic Concentration of the Included Studies A notable concern is the geographic concentration of the included studies. 75% are from Switzerland, the Netherlands, and Germany, with nearly half from Switzerland alone. Given Switzerland’s strong cultural and legal support for VSED, this limits the generalizability of the findings. The manuscript’s tone may give the impression that VSED is broadly accepted by healthcare professionals, but this does not reflect the reality in many countries. For example, in the U.S., where I have presented on VSED multiple times, reactions are typically marked by concern and unfamiliarity. While there is growing interest reflected in case reports, expert consensus, and opinion pieces in the U.S., these sources were not included in this review. As a result, the current synthesis may overstate acceptance of VSED globally. The authors mention geographic limitations in the discussion, but I recommend more prominently addressing this issue in other sections, particularly in findings such as #8 (“Advocacy by healthcare professionals”) where 7 of 9 studies are from Switzerland. It would be helpful to include or adapt the country-of-origin table from their previous review (Table 4 in the 2024 article) in this manuscript as well.
|
||||||||||||||||
Response 1: Thank you very much for this point of discussion. We absolutely agree, that the content of the review findings may be overrated. However, this is addressed by assessing the confidence into these review findings. This is recommended for systematic reviews. In Evidence based medicine it is common to use GRADE. We used GRADE-CERQual for Qualitative Evidence Synthesis accordingly. That means, that the result (here the review findings) “can be overrated” as they reflect only the underlying data. This is appropriate as long as the confidence into the result is assessed and was made transparent. We did this in a structured and transparent way (see appendix). The result is Table 3.
Nevertheless, we recognize, that this may be easily overlooked and made some changes. · we inserted the information “Assessment of confidence in review finding no. X: Low, very low, Moderate confidence – see Table 3.” after every review finding. · we integrated a table as you have recommended. Thank you for this advice ((Table 4, page 11, line 496-499). · we discussed the “geographical limitations” more deeply in the Results Section / 3.2. (page 10, line 470 - 478):
[“2. Geographical limitations: The main concerns related to the component of “relevance.” There were no geographical restrictions for the systematic search and, thus, for the research question. However, studies from only six countries were identified (Table 4), primarily from Switzerland (50%). Most identified studies (75%) are from Europe, rising to 100% when focusing on quantitative studies only. This raises serious concerns about generalizing the results to a global context (partial relevance) and the results (nos. 1-10) have to be interpreted cautiously. Accordingly, this was the main reason for downgrading the confidence in the CERQual approach (Table 3).”]
· we discussed the “geographical limitations” more deeply in the Discussion Section (page 17, line 624-646, part of the section “4.4. Confidence Assessment with CERQual and geographical limitation:
[“4.4 Confidence Assessments with CERQual and geographical limitation In general, the assessment of the findings on “attitudes towards VSED” (Findings nos. 5 and 6) gives more cause for concern than the findings dealing with caring, human solidarity, family cohesion or grief as fundamental human issues (Findings nos. 8, 9 and 10). Attitudes are assumed to be more dependent on culture and religious background than the fundamental human issues. Stängle et al. revealed that attitudes towards VSED vary among different linguistic and cultural regions within Switzerland. Other empirical data indicate a link between attitudes and faith [61,62,68,73]. It is likely that attitudes towards VSED differ between countries where religion plays a major role and those that are more secular. In countries where talking openly about dying or end-of-life options is considered taboo, VSED may be viewed more critically. The majority of identified studies (75%) originate from Europe, specifically Switzerland (4 studies), the Netherlands (2 studies), and Germany (2 studies). Two studies originated from the U.S. and one from New Zealand. In these countries and in nine U.S. states, medical aid in dying is legal or not explicitly prohibited [77]. Notably, euthanasia is legal in the Netherlands and New Zealand. Furthermore, in the Netherlands and Switzerland, right-to-die organizations have a long tradition, and end-of-life options are discussed more openly in public discourse than in other countries. These circumstances are likely to influence the attitudes of the study participants and may contribute to an overall trend towards more positive opinions. Therefore, we presume that future studies from countries with a more critical perspective could alter the findings on “attitudes towards VSED”. Notably, in addition to positive findings, burdensome and challenging experiences of healthcare professionals and relatives were also reported from these countries [48].
|
||||||||||||||||
Comments 2: Comment on Classification of VSED as Natural Death: For reference, see Uemura T, et al. J Am Med Dir Assoc. 2023;24:1442–1446, which illustrates the complexity and variation in how VSED is understood in different legal and clinical contexts. The manuscript would benefit from clarifying this distinction more explicitly and avoiding language that could be misinterpreted as making ethical or legal claims. |
||||||||||||||||
Response 2: Thank you very much for making us aware that the “wording” is inappropriate. This is a helpful advice. We have modified it accordingly to clarify that:
· First, we changed the wording in the Review Finding no. 6. (page 8) in the subheading and in the text: o Classification of VSED…à Attitudes towards VSED…(page 8, line 324,325,337) o Classification of whether VSED…à Assessment of whether VSED…(page 8, line 337) in the Review Finding no. 7 (Acceptance and its influencing factors) (page 8) o classify VSED à regard VSED (page 8, line 357 and line 358) · Second, we mentioned that in the Discussion Section within a new paragraph with the subheading 4.2. VSED as “natural death” or “suicide” (page 15, line 557 - 561):
[“The healthcare professionals` attitudes towards VSED as a natural death or suicide varies (Finding no. 6). Over half of the surveyed professionals consider VSED a natural death, while about one-quarter view VSED as equivalent to “letting die,” which is closest to the term of “passive euthanasia” [17]. A minority categorizes it as suicide. Since quantitative data derive from one Swiss and one German survey, final conclusions about quantifications must be drawn carefully. However, qualitative data from Switzerland, Germany, and the U.S. support different aspects of this finding. Notably, these attitudes towards VSED as natural death or suicide reflect the healthcare professionals’ perceptions and must be distinguished from the legal and normative classifications. Nevertheless, empirical data on healthcare professionals’ perspectives are invaluable for informing and supporting this ethical and legal debate.”] Factors that may influence these views include the patient's age and, probably the most decisive factor, their life expectancy (finding No. 6). The study by Stängle et al. indicates that participants with VSED experience are more likely to see VSED as natural death and less often as suicide compared to participants without VSED experience [17]. Participants in a U.S. study found it challenging to distinguish between deaths by suicide and deaths that were intentionally hastened but not considered suicide, noting that the boundaries between these categories are blurred [66]. Participants from the same study described “suicide” as a loaded term, associated with religious connotations, painful emotions and meanings, and regarded it as a “huge barrier” [66]. The assessment of whether VSED is considered suicide has practical implications for healthcare professionals. Viewing VSED as suicide can lead to its rejection by healthcare professionals and institutions [61,64,66,68]. Both may experience discomfort when deciding on a VSED accompaniment. Depending on the country's legislation, fear and uncertainties may arise concerning the legal situation and whether the accompaniment of VSED is legal or has to be regarded as a form of medically assisted suicide [66,68,69]. Ideally, there is a consistent attitude within institutions and teams regarding VSED and its assessment as natural death or suicide. Discrepant attitudes can create a dilemma for healthcare professionals between personal values and professional obligations. There are examples of institutions prohibiting accompaniment during VSED because they consider VSED to be suicide. In these cases, healthcare professionals who felt committed to patients' wishes found themselves compelled to act contrary to the institution's decision [66,68]. Moreover, physicians are confronted with difficulties in correctly completing the death certificate, as they must decide whether the death should be rated as "natural" or "unnatural“. Choosing "unnatural" may have problematic legal or regulatory consequences [22,28,29,74]. To date, there is no consensus among experts regarding the classification of VSED. Nevertheless, guidance is necessary to support physicians in addressing this issue, and solutions have been proposed [28,74].
|
||||||||||||||||
Comments 3: Comment on Study Period and Recent Literature: Even if a formal update of the systematic search is beyond the scope of this secondary analysis, acknowledging the potential for missed recent studies in the limitations section would strengthen the manuscript’s transparency and relevance.
|
||||||||||||||||
Response 3: Thank you for pointing that out. We have revised this accordingly in the section 4.6. Strengths and Limitations: page 18, line 720-722.
[“As outlined in Section 4.5, five additional empirical studies [78–82] have been published since our systematic search in 2021. This constitutes a limitation for the results of our analysis.“]
|
||||||||||||||||
Comments 4: Comment on Interpretation of “High Symptom Burden”: This distinction is important. Framing VSED as a response to symptoms that are potentially manageable through palliative care risks misrepresenting the role of existential distress, loss of autonomy, or perceived life fatigue, which may be more central motivators. The authors cite references 11 and 71 (both in German) as support for pain being a leading reason, but based on the abstract of ref 11, fatigue and loss of function appear more prominent, and pain is not mentioned. I encourage the authors to revisit this framing and distinguish more clearly between physical symptoms and broader psychosocial or existential factors. This clarification is crucial, particularly for readers seeking to understand when and why VSED may be considered over other palliative options. |
||||||||||||||||
Response 4: Thank you very much for pointing this out. · We have checked again the underlying data and came up with the same results about “pain”. Please find below the extracted and translated table 1 for line “pain” form the Stängle publication for your information:
Table 1 in [11] Stängle et al. 2019c:
· For more clarity, we have changed the order in the paragraph and followed your suggestion to distinguish better between physical and psychosocial/existential factors. Following the revised review finding: Page 6, line 219-231.
Motives - high symptom burden and suffering (reported by HCPs and relatives): Persons who have decided for VSED typically had a high symptom burden of physical and psychological symptoms, were in poor health or had an accumulation of age-related health problems [10,11,65,70,71]. Most reported physical symptoms include pain [11,71], fatigue and exhaustion, which have both somatic and psychic dimensions [10,11,71]. Psychosocial, spiritual or existential factors accounting for 21%–57% of reported motives [10,11], are described as “emerging life fatigue,” “meaninglessness of life,” “tired of living,” “missing a purpose of life,” “not wanting anymore,” or “loss of dignity” [10,11,64]. This led to a reduced quality of life, deterioration in health [10,11,62,65,71], and severe suffering [60,61,64,65,68] with no hope of improvement [10,11] (“I think, she was not tired of life but simply tired of suffering” [68]). The avoidance of suffering can also refer to the future and foreseeable health deterioration as described for early dementia [61].
For a better overview, what was changed and what was rearranged, please refer to the revised manuscript: Page 6, line 219-231. |
||||||||||||||||
Comments 5 and Respons 5: Minor comments: Authors: Changed: “The acceptance of a VSED decision depends on several factors. ”(line 343)
Authors: Changed: “Even though the data analyzed here reveal that healthcare professionals.”(line 591)
Authors: Changed: “Despite the acceptance of VSED,….”(line 594-5)
Authors: Changed: “Additionally, the approval rate of healthcare professionals...”(line 606)
Authors: We just deleted “far more” in the senctence. This sentence is saying that quantitative and qualitative research is strengthening the scientific rigor which we do not consider as a biased statement. There is bias, however we rated the confidence in the evidence. This is part of our methodology. Evidence from case reports is far more at risk in regards of selection bias. We integrate your concerns about the generalizability at different places in the manuscript (please refer to the other comments)
Line: 464-473: As I commented, the authors' acknowledgment of geographic limitations in this section is too understated. While they note that attitudes toward VSED may differ across countries, the manuscript does not adequately emphasize the significant concentration of included studies from regions where VSED is relatively well accepted, particularly Switzerland and the Netherlands. To strengthen transparency and improve interpretation of the findings, I recommend that the authors explicitly state how many studies originated from these countries and highlight how this may bias the results toward more favorable attitudes.
Authors Reply: · we discussed the “geographical limitations” more deeply in the Discussion Section (page 17, line 624-646, part of the section “4.4. Confidence Assessment with CERQual and geographical limitation - see also response to Comment 1)
Authors Reply: · this section was revised and is now integrated in the following paragraph: Discussion Section (page 17, line 624-646, part of the section “4.4. Confidence Assessment with CERQual and geographical limitation - see also response to Comment 1)
|
Reviewer 2 Report
Comments and Suggestions for Authors
Dear Authors,
Thank you for providing me with the opportunity to read this interesting paper. Below, I have listed my comments:
1) The introduction could briefly state why VSED has growing clinical or societal relevance today (e.g., increasing patient autonomy, aging populations, or debates around assisted dying laws). This would strengthen the rationale for the study.
2) The distinctions between the terms are not fully elaborated. For instance, how does voluntary palliated starvation differ from VSED in practice or intent? Clarifying this could enhance conceptual clarity and show deeper engagement with the literature.
3) In the methods, there is no indication of the keywords used.
4) Since the PRISMA guidelines were used, a flow diagram could have been added in the methodology.
5) In the findings, although findings are based on participant data, direct quotes or paraphrased reflections are largely absent. Including key illustrative quotes could strengthen authenticity and reader engagement.
6) In the discussion, while the paper mentions that some healthcare professionals view VSED as suicide, it doesn’t deeply explore why that view is held or its ethical implications. The controversial aspects of VSED are somewhat underdeveloped.
I hope this feedbakc is helpful
Author Response
Response to Reviewer 2 Comments
|
||
1. Summary |
|
|
Thank you very much for taking the time to review this manuscript. Please find the detailed responses below and the corresponding revisions/corrections highlighted/in track changes in the re-submitted files. |
||
2. Questions for General Evaluation |
Reviewer’s Evaluation |
Response and Revisions |
Does the introduction provide sufficient background and include all relevant references? |
Yes/Can be improved/Must be improved/Not applicable |
Please find our answers below |
Are all the cited references relevant to the research? |
Yes/Can be improved/Must be improved/Not applicable |
|
Is the research design appropriate? |
Yes/Can be improved/Must be improved/Not applicable |
|
Are the methods adequately described? |
Yes/Can be improved/Must be improved/Not applicable |
Please find our answers below |
Are the results clearly presented? |
Yes/Can be improved/Must be improved/Not applicable |
Please find our answers below |
Are the conclusions supported by the results? |
Yes/Can be improved/Must be improved/Not applicable |
Please find our answers below |
3. Point-by-point response to Comments and Suggestions for Authors |
||
Comments 1: The introduction could briefly state why VSED has growing clinical or societal relevance today (e.g., increasing patient autonomy, aging populations, or debates around assisted dying laws). This would strengthen the rationale for the study. |
||
Response 1: Thank you very much. Yes, we agree. We have revised this accordingly and inserted in the introduction (page 2, line 49 - 56 the following [“Aging populations have led to more frequent encounters with chronic and terminal illnesses, prompting discussions about dignified death. Additionally, self-determination and autonomy are increasingly recognized as significant societal trends in many countries influencing end-of-life decision-making. Ongoing debates concerning medically assisted dying laws further underscore the relevance on these end-of-life decisions. In this context, voluntary stopping of eating and drinking (VSED) has become an emerging focus of public and scientific attention as patients and healthcare professionals have to navigate complex ethical and legal landscapes.]
|
||
Comments 2: The distinctions between the terms are not fully elaborated. For instance, how does voluntary palliated starvation differ from VSED in practice or intent? Clarifying this could enhance conceptual clarity and show deeper engagement with the literature. |
||
Response 2: Thank you for pointing this out. We have modified it accordingly to clarify this point more deeply: Introduction, page 2 line 60-64 (here in red).
In addition to VSED, other terms include voluntary refusal of food and fluids (VRFF) [2], patient refusal of hydration and nutrition (PRHN) [3], voluntary palliated starvation [4] or terminal dehydration [5]. There may be slightly different definitions between these terms. For example, [Savulescu argues that every competent person has the right to refuse food and drink, even if this leads to death. Considering that every person is entitled to medical relief of their suffering through palliative care, which may include sedation and analgesia, he refers to this combination of refusal of food and drink and palliative symptom control as “voluntary palliative starvation” [4]]. Terminal dehydration has only been described in the context of terminally ill patients and includes the decision to forego artificial nutrition and hydration [5].
|
||
Comments 3: In the methods, there is no indication of the keywords used. |
||
Response 3: Thank you for pointing this out. This is the second publication on results of our systematic review. To avoid copyright issues, the PRISMA flowchart and the keywords are not published here again. But reference has been made at the end of the Materials and Methods section. As this may be easily overlooked we shifted now this paragraph to the beginning of the Materials and Methods section: page 4, line 124 - 129:
[This systematic review follows the PRISMA [58] and EPOC [59] guidelines. A protocol was registered in PROSPERO (CRD42022283743; https://www.crd.york.ac.uk/PROSPERO/view/CRD42022283743) on January 8, 2022. The PRISMA flowchart, the search strategy including search terms and a more detailed description of the methods on data analysis, quality assessment and confidence assessment are published in Mensger et al. 2024 and the respective Supplementary material [48].]
Additionally, we have inserted a note in the beginning of the Appendix: [“Please note, the PRISMA flowchart and the complete search strategy including search terms (text words and MeSH terms) are published in Mensger et al. 2024 and the respective Supplementary material [48].”]
|
||
Comments 4: Since the PRISMA guidelines were used, a flow diagram could have been added in the methodology. |
||
Response 4: Thank you for pointing this out. Please see our reply to Comment 3. Thank you very much.
|
||
Comments 5: In the findings, although findings are based on participant data, direct quotes or paraphrased reflections are largely absent. Including key illustrative quotes could strengthen authenticity and reader engagement. |
||
Response 5: Thank you very much for this valuable suggestion. We agree and inserted some direct quotes in the following Review Findings:
3. Self-determination and autonomy (page 7, line 269 - 271 and line 275-276): [“The reasons for VSED are linked to a strong expression of self-determination [62,64]. Frequently reported motives relate to autonomy and independence and the fear of losing these during the disease or in the future (57%–60% of the motives) (“… that this would at least mean a loss of independence. And for my mother [...] that was something impossible: loss of independence, that was inconceivable for her.” [65]) [10,11,64,65,71]. Maintaining control over the dying process is intended [61,65,66,68], and is seen as the last opportunity to express one’s will [64]. Accordingly, individuals opting for VSED are described as strong-minded, exhibiting a typical personality trait, as reported primarily by relatives [60,61,64,68]. Moreover, the course of VSED itself demands such will and courage (“[…] braver than asking for someone to give them a fatal injection.” [69]).”
4. VSED as the “best available option” or “better than other options” (page 7, line 287-289): […] [“VSED was also chosen when Medical Aid in Dying (MAiD) or Death with Dignity (DWD) was not available. This could be for legal reasons (not allowed in the country/state), for medical reasons (if the request was rejected because the person was not terminally ill) or if these options were not available for other reasons (e.g. cost in the U.S.) [10,60,61,66,67,70] (“She knew it was not an option, to give her medication to make her die. She wasn’t terminal by any stretch of the imagination. So I don’t think it was the ideal for her, but she was glad to have an exit strategy.” [60]).” […]
7. Acceptance and its influencing factors (page 8, line 352 - 353): [“Acceptance of the VSED decision is primarily related to the person's state of health [13,60–62,68,69,73]. The more visibly ill a person is and apparent the current symptom burden, the more understandable and acceptable the decision for VSED. The situation of terminally ill patients who “hasten it a little” [61] was primarily understood and accepted [13,61,69]. When persons are in good health or not visibly ill, for instance, when deterioration is expected in the future (e.g., early dementia) [13,60], the decision for VSED is less accepted [13,60,62,68,73] (“The healthier the patient, the more difficult” [73]).” […]
8. Advocacy by healthcare professionals (page 9, line 396 - 398): [“If the nurses had been caring for the patient for years and the decision was in line with the patient's personality and the course of the illness, it was obvious for them to accompany the person [68]. Once the decision had been made to accompany them, healthcare professionals could feel obliged to support the patient's wish to die through the VSED and can evolve into advocates for the person who wishes to die. They are committed to the patient's needs and do everything they can to provide the person with the support and environment they need [61,68,73] (“That is my task just as much as supporting or helping him or offering him services. That I also stand up for him and stand up for him and say: that is his wish (...) actually almost as an advocate.” [73]).”] […]
9. Advocacy by relatives (page 9, line 411 and 419 - 420): […] [“Once they have accepted the wish to die by VSED, relatives usually agree to accompany their loved ones [61,62,68]. Even in cases of rejection or lack of understanding, relatives believe that people have the right to choose their way of dying [60,62]. Once involved, relatives often play an active role and act as the patient's representative (“I promised her that I would fight to the death. Fight to her will!” [62]). They speak for and on behalf of the patient, are taking the lead in contacting physicians or hospices, and are making logistical arrangements to achieve a successful VSED [60,61,62]. They coordinate and organize farewells to family and friends and do everything essential for the patient [60,61]. They try to reduce contact with tempting food and drink [60] or monitor almost around the clock to ensure that patients are not given drinks or food against their will or forced to eat and drink [62]. They do all this as long as it is necessary and despite their insecurities, grief or other obstacles, such as rejection and accusations from others [61,62] (“And I just had to look at their faces. They clearly showed: ‘Do you want to kill your mother?’ And that's very depressing.” [62]).”] […]
10. Grieving process after VSED accompaniment (page 10, line 441-442 and 444-446): […] [“Nevertheless, how relatives perceive the grieving period depends on how they assess their accompanying and the dying situation. They can feel fulfilled, satisfied, and grateful if they rate both positive [62] (“These were the most beautiful fifteen days in my life with my mother. Yes. Because I could do anything for her.” [62]). However, if the dying process was not perceived as peaceful, for example, if the person was suffering or the death was agonizing, this made the grieving process more difficult [62,65] ("I just keep seeing this picture from the last three days. It just wasn't good. It's not a good way to die [...] If you ask me, I say perish" [62]).”] […]
|
||
Comments 6: In the discussion, while the paper mentions that some healthcare professionals view VSED as suicide, it doesn’t deeply explore why that view is held or its ethical implications. The controversial aspects of VSED are somewhat underdeveloped. |
||
Response 6: Thank you for pointing that out. We absolutely agree. We have inserted a paragraph in the Discussion section to discuss and explore that point more deeply (page 15, line 550 - 588: [“4.2 Voluntary stopping of eating and drinking as “natural death” or “suicide” The healthcare professionals` attitudes towards VSED as a natural death or suicide varies (Finding no. 6). Over half of the surveyed professionals consider VSED a natural death, while about one-quarter view VSED as equivalent to “letting die,” which is closest to the term of “passive euthanasia” [17]. A minority categorizes it as suicide. Since quantitative data derive from one Swiss and one German survey, final conclusions about quantifications must be drawn carefully. However, qualitative data from Switzerland, Germany, and the U.S. support different aspects of this finding. Notably, these attitudes towards VSED as natural death or suicide reflect the healthcare professionals’ perceptions and must be distinguished from the legal and normative classifications. Nevertheless, empirical data on healthcare professionals’ perspectives are invaluable for informing and supporting this ethical and legal debate. Factors that may influence these views include the patient's age and, probably the most decisive factor, their life expectancy (finding No. 6). The study by Stängle et al. indicates that participants with VSED experience are more likely to see VSED as natural death and less often as suicide compared to participants without VSED experience [17]. Participants in a U.S. study found it challenging to distinguish between deaths by suicide and deaths that were intentionally hastened but not considered suicide, noting that the boundaries between these categories are blurred [66]. Participants from the same study described “suicide” as a loaded term, associated with religious connotations, painful emotions and meanings, and regarded it as a “huge barrier” [66]. The assessment of whether VSED is considered suicide has practical implications for healthcare professionals. Viewing VSED as suicide can lead to its rejection by healthcare professionals and institutions [61,64,66,68]. Both may experience discomfort when deciding on a VSED accompaniment. Depending on the country's legislation, fear and uncertainties may arise concerning the legal situation and whether the accompaniment of VSED is legal or has to be regarded as a form of medically assisted suicide [66,68,69]. Ideally, there is a consistent attitude within institutions and teams regarding VSED and its assessment as natural death or suicide. Discrepant attitudes can create a dilemma for healthcare professionals between personal values and professional obligations. There are examples of institutions prohibiting accompaniment during VSED because they consider VSED to be suicide. In these cases, healthcare professionals who felt committed to patients' wishes found themselves compelled to act contrary to the institution's decision [66,68]. Moreover, physicians are confronted with difficulties in correctly completing the death certificate, as they must decide whether the death should be rated as "natural" or "unnatural“. Choosing "unnatural" may have problematic legal or regulatory consequences [22,28,29,74]. To date, there is no consensus among experts regarding the classification of VSED. Nevertheless, guidance is necessary to support physicians in addressing this issue, and solutions have been proposed [28,74].
|
Round 2
Reviewer 1 Report
Comments and Suggestions for Authors
Thank you for thoroughly addressing the concerns I previously raised. I enjoyed reading the revised manuscript and appreciated the thoughtful improvements. The addition of the table summarizing the country of origin of the referenced articles is particularly valuable. I hope this work sparks broader interest and engagement in this important topic. Thank you again.
Reviewer 2 Report
Comments and Suggestions for Authors
Thank you for revising the manuscript.